# RAG over Tables: Hierarchical Memory Index, Multi-Stage Retrieval, and Benchmarking

## Abstract

Retrieval-Augmented Generation (RAG) enhances Large Language Models (LLMs) by integrating them with an external knowledge base to improve the answer relevance and accuracy. In real-world scenarios, beyond pure text, a substantial amount of knowledge is stored in tables, and user questions often require retrieving answers that are distributed across multiple tables. Retrieving knowledge from a table corpora (i.e., various individual tables) for a question remains nascent, at least, for (1) how to understand intra- and inter-table knowledge effectively, (2) how to filter unnecessary tables and how to retrieve the most relevant tables efficiently, (3) how to prompt LLMs to infer over the retrieval, (4) how to evaluate the corresponding performance in a realistic setting. Facing the above challenges, in this paper, we first propose a table-corpora-aware RAG framework, named **T-RAG**, which consists of the hierarchical memory index, multi-stage retrieval, and graph-aware prompting for effective and efficient table knowledge retrieval and inference. Further, we first develop a multi-table question answering benchmark named **MultiTableQA**, which spans 3 different task types, 57,193 tables, and 23,758 questions in total, and the sources are all from real-world scenarios. Based on MultiTableQA, we did the holistic comparison over table retrieval methods, RAG methods, and table-to-graph representation learning methods, where T-RAG shows the leading accuracy, recall, and running time performance. Also, under T-RAG we evaluate the inference ability upgrade of different LLMs.

## 1 Introduction

Retrieve-Augmented Generation (RAG) has emerged as an effective approach to integrate external knowledge into Large Language Models (LLMs) to answer questions more accurately (Gao et al., 2023). In other words, inputting the retrieved external information into LLM's context window, RAG enhances LLM's inference capability and mitigates issues related to factual inaccuracies and hallucinations during reasoning and generation (Mei et al., 2025). For better retrieval performance in terms of effectiveness and efficiency, graph and hierarchical structures have recently been brought into RAG systems, and constructing these kinds of structures aims to organize and memorize the external knowledge base to retrieve the most relevant information in an efficient manner (Peng et al., 2024; Sarthi et al., 2024; Edge et al., 2024a; Gutierrez et al., 2024; Gutiérrez et al., 2025; Li et al., 2025; Han et al., 2025).

If not all, most of the recent RAG systems mainly focuses on text-based document external knowledge, while in the real world, a substantial amount of information and knowledge is stored in tables, which are frequently encountered in web pages, Wikipedia, and relational databases for various applications like question answering, personalized recommendation, molecular prediction (Fey et al., 2024; Dwivedi et al., 2025). However, retrieving the knowledge from tables for LLM inference remains nascent in current RAG system developments. To be specific, at least, there are four challenges that need to be solved. (1) Beyond the pure text, the information format in tables is more complex, which has a text-based header and content, also structured in a table format (Fey et al., 2024; Dwivedi et al., 2025). Therefore, understanding the table without information loss is not trivial. Moreover, a real-world question can care about the information from multiple tables, and this question can usually be ad-hoc, which means the pre-defined sparse primary-foreign key is not sufficient to locate the answer.

Hence, as shown in Figure 1, facing such a table corpora (various individual or loosely-connected tables), how to understand intra-table and inter-table knowledge is an unsolved problem. (2) Further, given the understanding of the corpora, building the memory index for RAG systems is also challenging. On the one hand, the memory index should be hierarchical (Edge et al., 2024a; Gutiérrez et al., 2025) such that a large portion of irrelevant information can be filtered out at an early stage and the accurate and representative answer can be located. On the other hand, with this hierarchical memory index, the corresponding retrieval should be co-designed with multiple adaptive granularities. (3) After

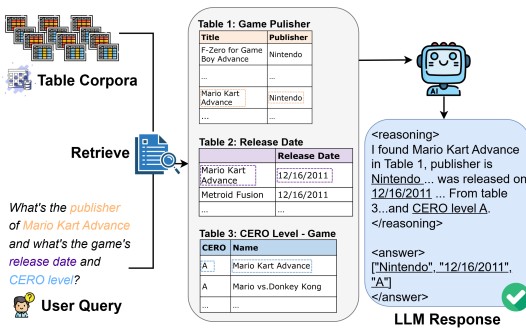

Figure 1: A Real-World Scenario for Cross-Table Question Answering.

the retrieval, a prompting engineering is necessary to organize the retrieved structural and textual information for LLM's inference (Mei et al., 2025). (4) Last but not least, we need a testbed to evaluate this realistic and cross-table RAG search, which is now still scarce (Seo et al., 2025; Yu et al., 2025), to the best of our knowledge.

Facing the above challenges, in this paper, we introduce a novel table-corpora-aware retrieval-augmented generation (RAG) framework, termed **T-RAG**. Our framework is carefully designed with three synergistic components: (1) a *hierarchical memory index* that organizes heterogeneous table knowledge at different granularities, (2) a *multi-stage retrieval* pipeline that progressively narrows the search space while preserving high recall, and (3) a *graph-aware prompting* mechanism that injects structured relational priors into LLMs. Together, these components enable more effective and efficient retrieval and reasoning over large-scale table corpora. Beyond methodology, we further establish the first large-scale multi-table question answering benchmark, denoted as **MultiTableQA**. The benchmark encompasses three representative task types, spanning 57,193 tables and 23,758 questions, all collected from realistic scenarios.

Comprehensive experiments on MultiTableQA deliver several key insights. We conduct holistic comparisons across three major paradigms: table retrieval methods, RAG methods, and table-to-graph representation learning methods. Results consistently demonstrate that T-RAG achieves state-of-the-art performance in terms of accuracy, recall, and runtime efficiency. Moreover, leveraging T-RAG, we systematically benchmark the reasoning and inference capabilities of frontier LLMs, including Qwen-2.5, Claude-3.5, and GPT-4o, showcasing the consistent performance gains brought by our framework across diverse model backbones.

## 2 PRELIMINARY

**Table Corpora**. Throughout the paper, a large-scale table corpora is a collection of tables, denoted as $\mathcal{T} = \{T_1, T_2, \ldots, T_t\}$. Each table $T$ comprises three components: *(i) Table Schema* $(A, H)$, which includes both the table caption $A$ and column headers $H$; *(ii) Table Entries* $E$, referring to the main body of the table values of $N$ rows and $M$ columns; and *(iii) Table Metadata* $D$, which provides additional description such as contextual details, associated resources, and representative example utterances. Formally, we represent each table as:

$$T = \{(A, H),\ E \in \mathbb{R}^{N \times M},\ D\},\ T \in \mathcal{T} \tag{1}$$

**Cross-table Retrieval for Question Answering Task**. Given the corpora $\mathcal{T}$, suppose we have a natural language question $q$ querying about the tables in $\mathcal{T}$ but cannot be directly answered by a pre-trained LLM. Then, let $\mathcal{M}$ denote a standard RAG pipeline that operates in two stages. First, $\mathcal{M}$ retrieves a subset of relevant tables from the corpus $\mathcal{T}$. Second, $\mathcal{M}$ generates an output $Y$ to augment downstream LLMs with additional context to answer question $q$. The overall RAG pipeline can be defined as $\mathcal{T}' = \text{Retrieve}_{\mathcal{M}}(\mathcal{T}, q)$ and $Y = \text{Generate}_{\mathcal{M}}(\mathcal{T}', q)$, where $\text{Retrieve}_{\mathcal{M}}(\cdot)$ selects top-$k$ most relevant tables, $\mathcal{T}'$ denotes the set of retrieved tables, and $\text{Generate}_{\mathcal{M}}(\cdot)$ produces a response $Y$ conditioned on both $\mathcal{T}'$ and $q$, as shown in Figure 1.

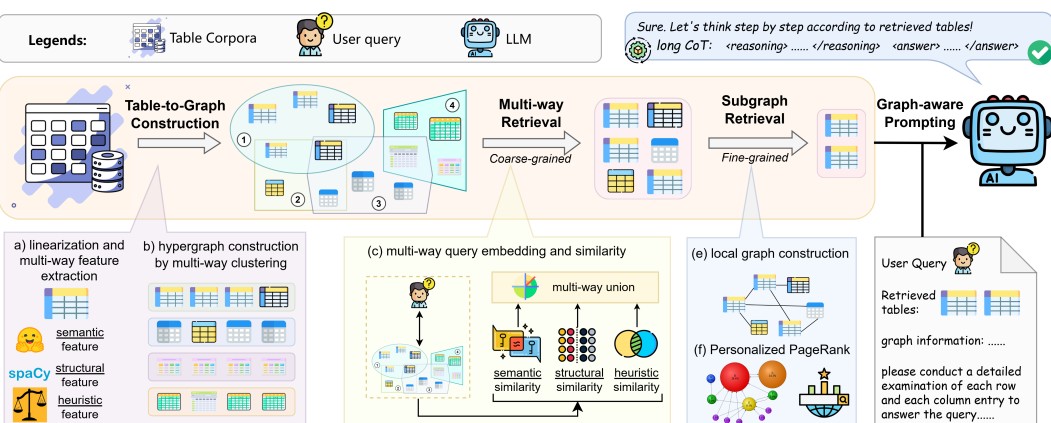

Figure 2: Overview of the T-RAG framework.

## 3 T-RAG: HIERARCHICAL MEMORY INDEX AND MULTI-STAGE RETRIEVAL

In general, T-RAG framework develops two core techniques, i.e., *hierarchical memory index* and *multi-stage retrieval*, to address the challenge of large-scale cross-table question answering. As illustrated in Figure 2, **first**, T-RAG regards each table as a potential node and organizes them into a heterogeneous hypergraph by clustering multi-way features of tables. To be specific, the heterogeneity stands for different types of hyperedges that originate from different focuses of clustering manners. **Second**, T-RAG employs the coarse-grained multi-way retrieval to select the optimal set of nodes (tables) in each cluster (i.e., hyperedge) corresponding to the query and instantiate selected tables as a subgraph. **Third**, T-RAG applies fine-grained subgraph retrieval to extract the final set of tables and employs a graph-aware prompting method for downstream LLMs' tabular reasoning. Next, we introduce these core techniques of T-RAG, and extra details are provided in Appendix A.

### 3.1 TABLE-TO-GRAPH CONSTRUCTION

In this section, T-RAG aims to organize the table corpora into a hypergraph, where a hyperedge connects a set of nodes (i.e., tables) collectively, not pairwise (Lee et al., 2024). The nodes inside a hyperedge are linked through membership in the same group, rather than direct one-to-one connections. In our setting, a hyperedge results from one individual clustering algorithm focusing on a part of the features of tables. Given a table that can have multi-way features and T-RAG can call different clustering algorithms, then the hyperedges are heterogeneous, i.e., different types.

**Table Linearization**. The first step is to linearize a table to capture both text structure and semantic properties. Specifically, given a table $T$, we extract its table schema components $(A, H)$ and concatenate them into a sequence as:

$$s = \left[ [\text{Table}], \bigoplus ([\text{Caption}], A), \bigoplus_{k=1}^{M} ([\text{Header}], h_k) \right], \quad (2)$$

where $\bigoplus$ denotes sequence concatenation and $h_k$ denotes the $k^{\text{th}}$ column header in $H$. Special tokens like [Table], [Caption], and [Header] are used to indicate structural positions within the table. In our implementation, we also experimented with alternative linearization methods, such as Table Summarization (Wang et al., 2022a). However, we observe that employing neural network models for linearization tends to disrupt the original table structure and increase computational complexity. Hence, we abandon these approaches in favor of the simple linearization method described above.

**Multi-way Feature Extraction**. Given every linearized sequence $s$, the next step is to compute three one-way feature vectors $\mathbf{x}^{(\text{sem})}, \mathbf{x}^{(\text{struct})}, \mathbf{x}^{(\text{heur})}$, in terms of semantics, structure, and heuristics, to maximally retain the original table information. Specifically, $\mathbf{x}^{(\text{sem})}$ is generated by a sequence-encoder (e.g., Sentence Transformer (Reimers, 2019) or Contriever (Izacard et al., 2021)) to capture the semantic content of the table. $\mathbf{x}^{(\text{struct})}$ is derived using spaCy to extract key format features—such

as token counts, part-of-speech (POS) tag frequencies, and punctuation counts—that effectively represent the table's structural properties. $\mathbf{x}^{(\text{heur})}$ is computed via heuristic methods, for instance, by employing a TF-IDF vectorizer, to capture the bag-of-words representation of the linearized table.

**Hypergraph Construction by Multi-way Clustering**. Now, T-RAG can finally construct a heterogeneous hypergraph $\mathcal{G} = (\mathcal{V}, \mathcal{E})$ that integrates the diverse features extracted from the linearized tables. With $t$ total number of tables in the table corpora, the node set is defined as: $\mathcal{V} = \{s_1, s_2, ..., s_t\}$, where each node $s_i$ is associated with its composite feature representation $\mathbf{x}_{s_i} = \left( \mathbf{x}_{s_i}^{(\text{sem})}, \mathbf{x}_{s_i}^{(\text{struct})}, \mathbf{x}_{s_i}^{(\text{heur})} \right)$. To capture the relationships between nodes from different perspectives, we define a set of heterogeneous edges with each heterogeneity corresponds to a feature type. For each feature type $\phi \in \{\text{sem}, \text{struct}, \text{heur}\}$, we apply KMeans clustering to partition all nodes into $K$ clusters $\{C_1^{(\phi)}, ..., C_K^{(\phi)}\}$. Mathematically, for each cluster $C$ as a feature-specified hyperedge $e$, we have $e_j^{(\phi)} = \{s_i \in C_j^{(\phi)} \mid j = 1, ..., K\}$, and the heterogeneous hyperedge set $\mathcal{E}$ is denoted as $\mathcal{E} = \bigcup_{\phi \in \{\text{struct}, \text{heur}, \text{sem}\}} \left\{ e_j^{(\phi)} \right\}$.

After constructing the hypergraph representing the table corpora, T-RAG uses a multi-stage **coarse-to-fine** retrieval process to identify the most relevant nodes $s_i$, for each incoming query $q$.

## 3.2 COARSE-GRAINED MULTI-WAY RETRIEVAL

Towards a query, T-RAG first applies a coarse-grained retrieval to hierarchically filter out irrelevant table data step-by-step with high efficiency.

**Representative Score**. We first need to define the representative score of a node, which will be frequently used in later node-to-node and node-to-query feature representation comparisons. Formally, the representative score $\mathbb{S}_{\text{rep}}$ between nodes $a$ and $b$ (e.g., node $s_i$ and query[1] $q$) with corresponding feature representations $\mathbf{x}_a^{(\phi)}$ and $\mathbf{x}_b^{(\phi)}$ on feature type $\phi$ is defined as:

$$\mathbb{S}_{\text{rep}}^{(\phi)}(a, b) = \frac{\langle \mathbf{x}_a^{(\phi)}, \mathbf{x}_b^{(\phi)} \rangle}{\|\mathbf{x}_a^{(\phi)}\| \|\mathbf{x}_b^{(\phi)}\|} \tag{3}$$

**Typical Node Selection**. For the efficient coarse-grained retrieval, for each cluster(i.e., hyperedge), we extract a small subset of nodes that can best represent that cluster, denoted as $\mathcal{V}_{\text{typ}}^{(\phi)}$. Specifically, for each cluster $C_j^{(\phi)}$ corresponding to feature type $\phi$, we choose the top-$k$ nodes with the highest representative scores:

$$\mathcal{V}_{\text{typ}}^{(\phi)} = \text{top-}k \left\{ \mathbb{S}_{\text{rep}}^{(\phi)}(s_i, \mu_j) \mid s_i \in C_j^{(\phi)} \right\} \tag{4}$$

where $\mu_j$ is the centroid of cluster $C_j^{(\phi)}$. The selection of typical nodes largely reduces computational complexity by restricting query comparisons to prototypical tables rather than the entire table corpora.

**Query-Cluster Assignment**. At query time, a natural language question $q$ is embedded using the same feature extraction methods, yielding representations of $\mathbf{x}_q$. We compute the representative scores between the query and each node in $\mathcal{V}_{\text{typ}}^{(\phi)}$ to select the optimal cluster $C^{*(\phi)}$ for each feature type $\phi$, i.e.,

$$C^{*(\phi)} = \arg\max_{C_j^{(\phi)}} \left\{ \frac{1}{|\mathcal{V}_{\text{typ}}^{(\phi)}|} \sum_{s_i \in \mathcal{V}_{\text{typ}}^{(\phi)}} \mathbb{S}_{\text{rep}}^{(\phi)}(q, s_i) \right\} \tag{5}$$

The final multi-way optimal cluster is the union across all feature types:

$$C^* = \bigcup_{\phi \in \{\text{sem}, \text{struct}, \text{heur}\}} C^{*(\phi)} \tag{6}$$

We later demonstrate in experiments that using the multi-way unioned clusters greatly enhances retrieval accuracy while incurring only a minimal increase in retrieved table size.

---

[1]Without loss of generality, we suppose a query can also have multi-way features.

## 3.3 FINE-GRAINED SUBGRAPH RETRIEVAL

With roughly related nodes (tables) extracted, next T-RAG needs to carefully narrow down the scope for locating the answer for a query.

**Local Subgraph Construction**. After the extraction of the optimal cluster $C^*$ carrying a bunch of hyperedge-linked nodes, T-RAG is then to leverage this abstract **set-wise** connectivity among table nodes to instantiate a densely **pair-wise** connected local subgraph, i.e., $\mathcal{G}_{\text{local}} = (\mathcal{V}_{\text{local}}, \mathcal{E}_{\text{local}})$ and

$$\mathcal{V}_{\text{local}} = \{s_i \mid s_i \in C^*\}, \ \mathcal{E}_{\text{local}} = \left\{(s_i, s_j) \in C^* \times C^* \mid \mathbb{S}_{\text{rep}}^{(\text{sem})}(s_i, s_j) \geq \tau\right\} \tag{7}$$

where $\tau \in [0, 1]$ is a similarity threshold, and each edge is weighted by its corresponding representative score. Note that after the coarse-grained filtering stage, only semantic features are utilized.

**Iterative Personalized PageRank for Retrieval**. Given the local subgraph $\mathcal{G}_{\text{local}} = (\mathcal{V}_{\text{local}}, \mathcal{E}_{\text{local}})$, for fine-grained retrieval, we can first compute a similarity matrix $\mathbf{S}$ over the candidate nodes $\mathcal{V}_{\text{local}}$:

$$\mathbf{S}_{ij} = \begin{cases} \mathbb{S}_{\text{rep}}^{(\text{sem})}(s_i, s_j), & \text{if } (s_i, s_j) \in \mathcal{E}_{\text{local}}, \\ 0, & \text{otherwise.} \end{cases} \tag{8}$$

Then, we obtain the transition matrix $\mathbf{P}$ by row-normalizing $\mathbf{S}$. The personalization vector $\mathbf{h} \in \mathbb{R}^{t_{\text{local}}}$ is computed from the query $q$ as:

$$h_i = \frac{\mathbb{S}_{\text{rep}}^{(\text{sem})}(q, s_i)}{\sum_{j=1}^{t_{\text{local}}} \mathbb{S}_{\text{rep}}^{(\text{sem})}(q, s_j)} \tag{9}$$

Finally, we update the iterative personalized PageRank vector (Andersen et al., 2006) $\mathbf{v}$ for each iteration $\sigma$ by $\mathbf{v}^{(\sigma+1)} = (1 - \alpha)\mathbf{h} + \alpha \mathbf{P} \mathbf{v}^{(\sigma)}$ with the damping factor $\alpha \in (0, 1)$ and initialization $\mathbf{v}^{(0)} = \mathbf{h}$. The iteration continues until convergence, i.e., $\|\mathbf{v}^{(\sigma+1)} - \mathbf{v}^{(\sigma)}\|_1 < \epsilon$ for a small tolerance $\epsilon > 0$. The final PageRank score vector $\mathbf{v}$ ranks the nodes in $\mathcal{V}_{\text{local}}$. The top-ranked nodes are then selected as the final retrieved table nodes, denoted as $\mathcal{V}_{\text{final}}^*$.

## 3.4 GRAPH-AWARE PROMPTING

After obtaining the final set $\mathcal{V}_{\text{final}}^*$, we apply a graph-aware prompting method to enable downstream LLMs to effectively interpret the retrieved tables and perform tabular reasoning. Our prompt structure comprises two key components: (i) graph information insertion and (ii) instructions for hierarchical long chain-of-thought (CoT) generation. Due to space constraints, we provide a detailed description of our multi-step prompting strategy in Appendix A.2.

## 4 MULTITABLEQA: ONE QUESTION QUERYING MULTIPLE TABLES

In MultiTableQA, based on the different types of user queries, we define three cross-table tasks: (i) Table-based Fact Verification (TFV), (ii) Single-hop Table Question Answering (Single-hop TQA), and (iii) Multi-hop Table Question Answering (Multi-hop TQA). In general, the essential difference between single-hop and multi-hop is whether the answer(s) of the query are located in one or multiple table cells, though all tasks require cross-table reasoning to retrieve the final answer. Detailed definitions of each task are provided in Appendix B.2 with concrete question-answer examples. The overall benchmark construction is illustrated in Figure 3. Additional details are provided in Appendix B. Next, we introduce our efforts to obtain real-world multi-table source, questions, and answers.

Table 1: Summarized Statistics of MultiTableQA

| Task Type | #Tables | #Queries | #Avg Rows | #Avg Cols |
|---|---|---|---|---|
| TFV | 34,351 | 15,106 | 5.8 | 5.7 |
| Single-hop TQA | 17,229 | 6,106 | 7.4 | 4.5 |
| Multi-hop TQA | 5,523 | 2,573 | 13.8 | 7.3 |

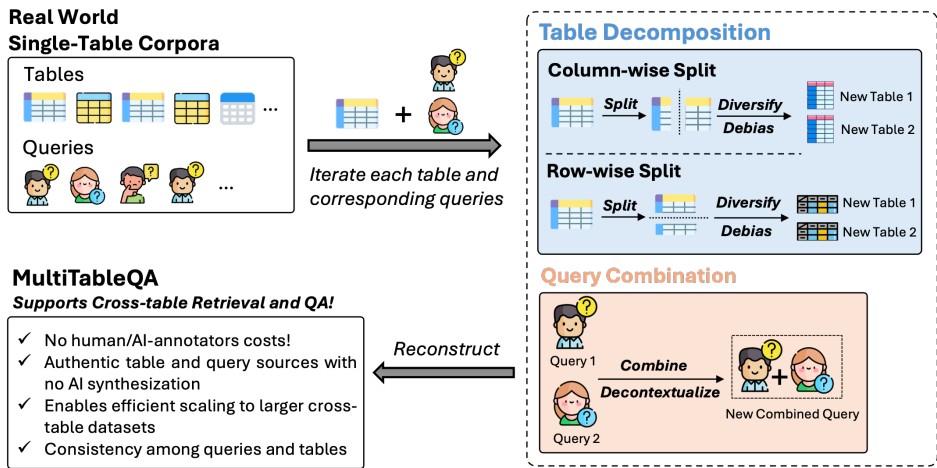

Figure 3: Illustration of the MultiTableQA benchmark construction pipeline.

## 4.1 SOURCE TABLE DECOMPOSITION.

**Table Sources**. MultiTableQA collects raw single table sources from real-world, human-annotated datasets, including HybridQA (Chen et al., 2020), SQA (Iyyer et al., 2017), Tabfact (Chen et al., 2019), and WikiTables (Pasupat and Liang, 2015). Overly simplified tables are filtered out, yielding a curated collection of 20k tables. We then apply the row-/column-wise splitting on these collected table sources and decompose them into 60k tables as our multi-table data.

**Row-/Column-wise Splitting**. We begin by pre-processing the original single-table data to filter out tables with small dimensions (specifically, those with $M \leq 3$ columns and $N \leq 3$ rows), as they are not suitable for decomposition. Then given a random $T \in \mathcal{T}$, we apply either row-wise or column-wise splitting.

- Row-wise Splitting: Formally, we divide the table entries $E$ into $n$ disjoint sets, i.e., $E = \bigcup_{i=1}^{n} E_i$ with $E_i \in \mathbb{R}^{N_i \times M}$ and $\sum_{i=1}^{n} N_i = N$. After processing table entries, we maintain each sub-table with the same table schema and metadata as the original table.

- Column-wise splitting: Given that relational tables often have a column-major logical structure, we retain the first column entry $c_1$ (typically the primary key or leading attribute) in all sub-tables. Then the original table entries can be rewritten as $E = [c_1, E']$. The rest of the entries $E'$ are decomposed into $m$ disjoint sets, i.e., $E' = \bigcup_{j=1}^{m} E'_j$ with $E'_j \in \mathbb{R}^{N \times M_j}$ and $\sum_{j=1}^{m} M_j = M - 1$. The overall table entries splitting column-wise are given by $E = \bigcup_{j=1}^{m} \left( [c_1, E'_j] \right)$. We then separate column headers corresponding to each sub-table entry and maintain other components the same as the original table.

**Debiasing and Diversification**. To mitigate potential biases in retrieval evaluation, particularly those introduced by sub-tables sharing identical content from the same root table, we apply a twofold debiasing strategy. First, we paraphrase the captions of all sub-tables derived from the same source table to reduce textual redundancy and semantic overlap. Second, we randomly permute the column or row order within each sub-table to encourage structural diversity. Third, we balance the number of sub-tables produced by row-wise and column-wise partitioning to ensure a fair distribution across different splitting strategies. This process enhances the robustness and diversity of the dataset without incurring additional human annotation costs or relying on model-generated synthetic data, which could introduce unintended evaluation biases.

**Retrieval Difficulty**. In MultiTableQA, we classify splitter tables into three difficulty levels based on the number of sub-tables derived from each source table: *(i) Easy:* No split, *(ii) Medium:* split into two sub-tables, and *(iii) Hard:* split into three sub-tables.

## 4.2 QUERY COMBINATION

To further increase the complexity of query retrieval and support multifaceted information needs requiring sequential reasoning, we adopt a simple yet effective composition strategy that integrates existing atomic queries into more complex, multi-step queries.

**Query Sources**. We first collect over 150k raw queries associated with the aforementioned single-table sources. Typically, each table is accompanied by 1 to 8 user queries. Importantly, these questions are sourced from real-world question-answering deployment systems or annotated by human experts (e.g., via the Amazon Mechanical Turk platform (Chen et al., 2020; 2019)), rather than being synthetically generated.

**Query Combination**. We then filter out vague and context-repetitive queries using common linguistic and context-aware heuristics (Cao et al., 2008; Van Rooij, 2011). Specifically, we apply techniques such as stopword ratio analysis, minimum query length thresholds, and similarity-based redundancy detection to refine the dataset. This process results in a curated set of over 80k high-quality queries. After that, for these multifaceted or sequential queries originally from the same single table, we utilize connecting words (e.g., "AND", "Furthermore", "Based on [previous query]") to merge them into a single, extended query. This process results in a final set of 25k combined queries.

**Query Decontextualization**. We observe that many original queries,especially those tied to a single table, often rely heavily on implicit contextual cues, rendering them unsuitable for standalone retrieval. To enhance clarity and ensure that queries are fully self-contained, we follow the decontextualization strategies proposed by Chen et al. (2020); Choi et al. (2021). Specifically, we replace ambiguous discourse markers and demonstrative pronouns with explicit noun phrases or entity references, resulting in clearer and context-independent queries that better support open-domain retrieval settings.

**Utilization**. In our experiments, we first randomly select 3,000 queries along with their ground-truth answers as "example queries" and pair them with their corresponding tables in the constructed table corpora. These selected queries are incorporated into the table metadata for data augmentation and example-based demonstration purposes. We then randomly select 1,000 queries for each of the three task types to form the testing set. The remaining 19k queries are set aside as the training set for future research.

## 5 EXPERIMENTS

In this section, we deploy our proposed framework on MultiTableQA. We demonstrate that T-RAG exhibits superior performance on both retrieval and downstream generation and reasoning. Full experimental setups are provided in Appendix C.

**Baselines**. To rigorously assess the effectiveness of our method, we compare each component of our proposed framework against a diverse set of baselines: (i) *Table Retrieval*, including DTR (Herzig et al., 2021), Table-Contriever (Izacard et al., 2021), Table-E5 (Wang et al., 2022b), and Table-LLaMA (Zhang et al., 2023a); (ii) *RAG*, including RALM (Ram et al., 2023) and ColBERT (Santhanam et al., 2021); (iii) *Table-to-Graph Representation*, including single feature extraction (Sec 3.1, and tabular representation learning models such as TAPAS (Herzig et al., 2020) and TaBERT (Yin et al., 2020). Detailed baseline descriptions are provided in Appendix C.1.

**Metrics**. For retriever evaluation, we employ Acc@$k$ and Recall@$k$ metrics with choices of $[10, 20, 50]$. For downstream LLMs tabular reasoning, we use Exact Match and F1 scores. We leave the model settings and additional experimental setups in Appendix C.2.

## 5.1 MAIN RESULTS

Table 2 presents the overall retrieval results for T-RAG and the baseline methods. To highlight, T-RAG achieves accuracy improvements ranging from 1.2% to 11.4% and recall gains from 1.5% to 12.5% when compared to table retrieval baselines. We also observe that traditional RAG-based methods perform poorly, indicating that relying solely on semantic similarity is insufficient to retrieve from tables. Compared with Table-to-Graph Representation baselines, T-RAG consistently yields performance gains. For example, T-RAG achieves improvements of up to 9.4% in recall@50 on TFV

Table 2: Main results of T-RAG and baseline methods across three tasks in MultiTableQA. We report average retrieval accuracy and recall (@10, 20, 50) on three runs. The best performance is **bold** and the second-best is underlined.

| Category | Methods | TFV | | | | | | Single-hop TQA | | | | | | Multi-hop TQA | | | | | |
|---|---|---|---|---|---|---|---|---|---|---|---|---|---|---|---|---|---|---|---|
| | | Accuracy | | | Recall | | | Accuracy | | | Recall | | | Accuracy | | | Recall | | |
| | | 10 | 20 | 50 | 10 | 20 | 50 | 10 | 20 | 50 | 10 | 20 | 50 | 10 | 20 | 50 | 10 | 20 | 50 |
| Table Retrieval | DTR | 21.1 | 27.8 | 36.2 | 36.4 | 43.0 | 51.4 | 35.8 | 46.5 | 59.5 | 46.8 | 56.3 | 67.7 | 38.9 | 46.4 | 57.8 | 44.2 | 51.5 | 62.0 |
| | Table-Contriever | 23.4 | 30.1 | 40.1 | 40.5 | 47.8 | 57.1 | 39.8 | 51.7 | 66.1 | 52.0 | 60.6 | 71.2 | 43.2 | 51.5 | 64.2 | 49.1 | 57.2 | 68.9 |
| | Table-E5 | 23.4 | 30.4 | 40.3 | 42.2 | 49.1 | 58.0 | 43.5 | 54.7 | 69.6 | **56.5** | **66.7** | 78.9 | 49.6 | 56.9 | 69.1 | 55.0 | 62.3 | 72.9 |
| | Table-LLaMA | 34.9 | 44.1 | 56.1 | 53.5 | 59.2 | 69.5 | 40.6 | 52.3 | 72.0 | 48.3 | 61.8 | 75.4 | 45.8 | 49.1 | 61.8 | 51.9 | 55.8 | 64.3 |
| RAG | RALM | 6.4 | 8.5 | 10.1 | 8.2 | 9.7 | 12.5 | 4.3 | 8.2 | 13.7 | 7.5 | 11.2 | 14.6 | 12.1 | 15.9 | 18.3 | 16.3 | 19.7 | 21.4 |
| | ColBERT | 14.9 | 18.3 | 22.1 | 20.8 | 26.4 | 31.6 | 16.8 | 23.6 | 36.4 | 17.3 | 23.9 | 36.4 | 36.9 | 47.6 | 58.3 | 39.3 | 46.5 | 61.1 |
| Table-to-Graph Representation | Single-head (heur) | 4.9 | 7.8 | 8.9 | 6.9 | 10.2 | 11.4 | 6.3 | 10.5 | 11.5 | 7.3 | 9.5 | 14.0 | 8.2 | 9.4 | 9.8 | 8.8 | 11.3 | 12.6 |
| | Single-head (struc) | 14.6 | 21.9 | 28.4 | 27.0 | 31.3 | 38.6 | 21.5 | 29.5 | 41.9 | 24.0 | 29.5 | 43.9 | 26.4 | 28.9 | 35.3 | 29.4 | 33.7 | 38.3 |
| | Single-head (sem) | 18.0 | 26.1 | 34.8 | 31.6 | 37.4 | 45.3 | 25.7 | 36.4 | 50.3 | 28.6 | 36.7 | 52.8 | 32.5 | 35.5 | 43.1 | 36.3 | 39.6 | 46.2 |
| | Lattice | 7.7 | 11.6 | 13.1 | 12.3 | 15.1 | 17.2 | 13.3 | 16.7 | 23.5 | 14.5 | 17.7 | 24.8 | 15.2 | 16.7 | 20.6 | 16.3 | 18.4 | 21.7 |
| | TaBERT | 33.8 | 45.5 | 51.6 | 48.5 | 57.8 | 63.9 | 41.6 | 52.9 | 74.4 | 45.8 | 56.2 | 78.9 | 48.0 | 52.9 | 65.6 | 51.6 | 58.2 | 66.9 |
| | TAPAS | 35.6 | **48.5** | 53.8 | 51.1 | 60.3 | 66.5 | 44.3 | 55.7 | 78.3 | 48.2 | 59.0 | 82.5 | 50.6 | 55.8 | 68.7 | 54.3 | 61.2 | 70.1 |
| | **T-RAG** | **36.1** | 47.9 | **59.4** | **55.9** | **64.3** | **75.9** | **47.3** | **62.9** | **83.1** | 51.5 | 63.3 | **86.8** | **57.2** | **60.3** | **72.7** | **62.5** | **67.6** | **76.8** |

and 8.2% in recall@10 on Multi-hop TQA. This underscores the importance of our overall graph construction and retrieval processes.

Table 3: Experimental results on downstream LLMs' cross-table QA performance using T-RAG. We also report the average improvement in downstream performance compared to the strongest corresponding baseline methods such as Table-E5 and TAPAS. Full experiment results are reported in Appendix D.1.

| Models | TFV | | | Single-hop TQA | | | | | | Multi-hop TQA | | | | | | Improv. |
|---|---|---|---|---|---|---|---|---|---|---|---|---|---|---|---|---|
| | EM@10 | EM@20 | EM@50 | EM@10 | F1@10 | EM@20 | F1@20 | EM@50 | F1@50 | EM@10 | F1@10 | EM@20 | F1@20 | EM@50 | F1@50 | (↑Δ) |
| Phi-3.5-mini | 22.3 | 45.9 | 44.3 | 26.2 | 28.1 | 27.0 | 28.5 | 25.2 | 26.5 | 13.9 | 14.2 | 15.6 | 16.7 | 11.8 | 12.6 | 16.9% |
| LLaMA-3.2-3B | 41.6 | 48.3 | 43.7 | 19.1 | 19.4 | 22.8 | 23.1 | 23.6 | 24.1 | 11.3 | 14.7 | 15.9 | 16.8 | 13.2 | 13.7 | 13.1% |
| Qwen-2.5-7B | 47.2 | 53.8 | 46.4 | 31.2 | 35.4 | 30.8 | 32.3 | 36.8 | 28.1 | 24.8 | 27.5 | 24.2 | 28.4 | 30.6 | 24.8 | 11.8% |
| LLaMA-3.1-8B | 48.1 | 52.7 | 50.9 | 33.2 | 34.1 | 32.6 | 33.6 | 31.2 | 32.4 | 24.7 | 25.6 | 26.2 | 26.6 | 28.8 | 27.4 | 11.9% |
| LLaMA-3.1-70B | 51.2 | 55.7 | 62.1 | 42.8 | 45.7 | 44.2 | 47.1 | 48.1 | 50.2 | 31.4 | 30.8 | 32.6 | 31.5 | 35.8 | 36.1 | 8.2% |
| Claude-3.5-Sonnet | 53.3 | 60.6 | 65.8 | 49.2 | 53.7 | 53.2 | 55.0 | 51.9 | 52.7 | 44.8 | 45.3 | 47.1 | 47.9 | 44.1 | 44.5 | 9.1% |
| GPT-4o-mini | 44.8 | 52.1 | 57.2 | 39.4 | 39.6 | 38.3 | 38.7 | 41.2 | 44.5 | 37.4 | 38.6 | 34.9 | 35.8 | 36.1 | 37.3 | 5.2% |
| GPT-4o | 52.7 | 63.1 | 66.5 | 48.9 | 52.6 | 50.4 | 53.6 | 52.1 | 56.6 | 41.7 | 42.9 | 45.3 | 46.8 | 49.6 | 50.9 | 13.6% |

**Downstream Performance.** In Table 3, we present the downstream tabular reasoning performance on MultiTableQA across LLMs with diverse types and parameter scales. The results show that T-RAG consistently improves cross-table question answering performance, yielding an average gain of 11.2% compared to the strongest baseline methods across all evaluated LLMs. We provide the full comparison with other baselines in Appendix D.1.

## 5.2 GENERALIZABILITY OF T-RAG

To further evaluate the robustness and generalizability of T-RAG beyond our constructed benchmark, we conduct additional experiments on the Spider dataset (Yu et al., 2018). In this setting, each query is grounded in a single table selected from a large pool of heterogeneous tables. We adapt T-RAG to this single-table scenario by treating the task as retrieving the most relevant target table from the entire table corpus. Results in Table 4 demonstrate that T-RAG generalizes well to additional datasets such as Spider, where each table inherently differs in structure and content. Further, we leave more ablation and parameter studies in Appendix D.2 and D.3.

## 5.3 EFFICIENCY ANALYSES

We evaluate the efficiency of T-RAG on the full MultiTableQA benchmark, consisting of 25k queries and 60k tables. Table 5 reports the end-to-end latency of each component in our framework across different task types. Graph construction and coarse-grained clustering are generally faster, while fine-grained PageRank-based retrieval incurs higher latency due to its more complex subgraph scoring and ranking operations. Despite incorporating a multi-stage retrieval process, GTA still achieves

Table 4: Performance of GTR on the Spider dataset. We report results using GPT-4o as the downstream LLM generator for question answering tasks.

| Spider | Accuracy@10 | Recall@10 | EM@10 |
|---|---|---|---|
| DTR | 14.9 | 21.7 | 63.3 |
| Table-E5 | 22.1 | 33.5 | 61.6 |
| TAPAS | 25.4 | 38.6 | 71.1 |
| T-RAG | **31.9** | **42.3** | **73.9** |

Table 5: Latency evaluation of each component in T-RAG, along with comparisons to baselines.

| Latency (min) | TFV | Single-hop TQA | Multi-hop TQA |
|---|---|---|---|
| Table-to-Graph | 0.4 | 0.2 | 0.1 |
| Coarse-grained | 24.0 | 11.8 | 11.2 |
| Fine-grained | 108.7 | 66.8 | 23.3 |
| T-RAG (Total) | 133.1 | 78.8 | 34.6 |
| TAPAS | 117.3 | 96.8 | 49.7 |
| Table-E5 | 215.8 | 134.2 | 69.2 |

Table 6: The number of tables remaining before retrieval, after coarse-grained multi-head retrieval, and after fine-grained subgraph retrieval. We set the parameters $K = 10$, $k = 100$, and $\mathcal{V}^*_{\text{final}} = 10$.

| Task | Before retrieval | After coarse-grained | After fine-grained |
|---|---|---|---|
| TFV | 34,351 | 4204 ($\downarrow 87.8\%$) | 10 ($\downarrow 99.9\%$) |
| Single-hop TQA | 17,229 | 2184 ($\downarrow 87.3\%$) | 10 ($\downarrow 99.9\%$) |
| Multi-hop TQA | 5,523 | 649 ($\downarrow 88.2\%$) | 10 ($\downarrow 99.8\%$) |

comparable overall latency compared to strong baselines such as TAPAS and Table-E5. We further report the retrieval efficiency in Table 6.

## 6  RELATED WORK

**Table Question-Answering**. Table Question Answering (Vakulenko and Savenkov, 2017) focuses on answering user queries by reasoning over tabular data. Existing benchmarks and datasets (Pasupat and Liang, 2015; Aly et al., 2021) have predominantly concentrated on single-table QA settings, with each query corresponding to a single table. Recently, several studies (Yu et al., 2018; Pal et al., 2023; Liu et al., 2023; Zhao et al., 2022) have focused on incorporating multiple tables for downstream tabular reasoning. More recent works, such as MT-RAIG (Seo et al., 2025) and MMQA (Wu et al.), incorporate a retrieval process to ask models to identify relevant tables with synthesized queries. **Graph-based RAG**. GraphRAG (Peng et al., 2024) extends traditional retrieval-augmented generation (RAG) by modeling inter-document relationships through graph structures, improving both retrieval and generation (Edge et al., 2024b). While prior works (Lewis et al., 2020; Wang et al., 2023b; Asai et al.; Siriwardhana et al., 2023; Xu et al., 2023; Zhang et al., 2023b) often assume pre-structured graphs (e.g., knowledge graphs) with explicit edge relations (Wang et al., 2023a; Agrawal et al., 2024), real-world data such as tables are typically semi-structured and lack explicit relational schema. Effectively applying GraphRAG to such data remains an open challenge.

## 7  CONCLUSION

In this paper, we present a comprehensive framework T-RAG to achieve multi-table retrieval across large-scale table corpora. Then, we first construct MultiTableQA as a novel multi-table benchmark by leveraging an adaptive dataset transformation approach. Experimental results on both retrieval and generation processes demonstrate the effectiveness and efficiency of our approach in enhancing cross-table retrieval and reasoning.

## REPRODUCIBILITY STATEMENT

In this paper, we present detailed implementation descriptions Appendix A and Appendix B. To ensure reproducibility, we commit to sharing the code with reviewers upon request during the review phase, and we will publicly release the full codebase along with the benchmark upon acceptance and publication of this work.

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

# Appendix

## A   ADDITIONAL DETAILS ON T-RAG

### A.1   REPRESENTATIVE SCORE INTERPRETATION

The formulation of Representative Score in Section 3.2 corresponds mathematically to cosine similarity; however, we refer to it as a representative score to highlight the specific role in our setting: quatify how well one node serves as a representative,semantically or structurally, for another under a given feature type. Unlike traditional use cases of cosine similarity that merely assess directional alignment in a vector space, our use emphasizes task-specific interpretability, where high representative scores suggest strong mutual informativeness between subtables and queries or among subtables themselves.

### A.2   GRAPH-AWARE PROMPTING METHOD

After obtaining the final set of retrieved tables $\mathcal{V}^*_{\text{final}}$, we design graph-aware prompt inputs to enable downstream LLMs to effectively interpret the retrieved information and perform tabular reasoning. Specifically, our prompts emphasize the following key aspects.

**Graph Information Insertion.**   Upon extracting the final retrieved tables from the local sub-graph, we observe that incorporating graph-related information—such as representative scores among nodes—enhances downstream LLMs' ability to interpret the weighted relationships among the retrieved tables. Consequently, we embed node indices along with their corresponding inter-node weighted edge scores into the prompt.

**Hierarchical Long CoT Generation.**   Inspired by recent advances in long Chain-of-thought (Wei et al., 2022; Yeo et al., 2025) and test-time scaling (Jaech et al., 2024; Guo et al., 2025) where long chain-of-thought chains are generated alongside final answers, we employ a similar strategy for downstream tabular reasoning. Specifically, we prompt the LLMs to reason step-by-step by addressing the following: (i) identify the most relevant tables from the provided table set $\mathbf{V}^*_{\text{final}}$; (ii) elucidate the connection between the query and the selected tables; and (iii) conduct a detailed examination of each row and column entry to extract the information most pertinent to the query, ultimately arriving at the final answer. The outputs from the LLM are structured into two components: the reasoning process enclosed within the tags *<reasoning>* and *</reasoning>*, and the final answer enclosed within the tags *<answer>* and *</answer>*.

Table 7: Downstream results of LLMs tabular reasoning using the baseline method of Table-E5.

| Models | TFV | | | Single-hop TQA | | | | | | Multi-hop TQA | | | | | |
|---|---|---|---|---|---|---|---|---|---|---|---|---|---|---|---|
| | EM@10 | EM@20 | EM@50 | EM@10 | F1@10 | EM@20 | F1@20 | EM@50 | F1@50 | EM@10 | F1@10 | EM@20 | F1@20 | EM@50 | F1@50 |
| Phi-3.5-mini | 16.2 | 35.8 | 31.5 | 18.6 | 20.0 | 19.2 | 20.3 | 17.9 | 18.8 | 9.9 | 10.1 | 11.1 | 11.9 | 12.4 | 12.0 |
| LLaMA-3.2-3B | 36.9 | 41.8 | 35.2 | 13.9 | 14.1 | 16.5 | 16.8 | 17.1 | 17.5 | 8.2 | 10.7 | 11.5 | 12.2 | 9.6 | 9.9 |
| Qwen-2.5-7B | 41.2 | 43.7 | 36.9 | 30.3 | 32.2 | 27.2 | 29.3 | 31.8 | 30.5 | 22.5 | 24.9 | 21.9 | 25.7 | 27.7 | 22.5 |
| LLaMA-3.1-8B | 42.6 | 44.7 | 50.3 | 28.5 | 29.1 | 30.9 | 31.0 | 26.8 | 27.9 | 21.3 | 21.8 | 22.7 | 23.7 | 24.6 | 23.5 |
| LLaMA-3.1-70B | 40.3 | 46.2 | 58.1 | 36.1 | 39.2 | 37.9 | 40.9 | 41.7 | 42.5 | 26.5 | 26.7 | 28.3 | 27.7 | 30.2 | 31.4 |
| Claude-3.5-Sonnet | 46.2 | 51.0 | 54.8 | 38.6 | 43.2 | 39.9 | 43.8 | 42.7 | 45.4 | 34.1 | 38.7 | 36.4 | 37.9 | 39.3 | 41.2 |
| GPT-4o-mini | 40.3 | 46.7 | 48.3 | 34.3 | 34.5 | 33.4 | 33.6 | 36.0 | 38.8 | 32.5 | 33.9 | 30.6 | 31.1 | 31.5 | 32.8 |
| GPT-4o | 44.1 | 56.3 | 56.1 | 44.2 | 49.5 | 45.8 | 50.2 | 48.6 | 52.6 | 39.2 | 40.3 | 42.7 | 44.0 | 46.7 | 46.2 |

**Multi-step Prompting.**   In line with our graph-aware and long chain-of-thought generation strategies, our prompt design involves three steps: (i) highlighting graph-related information, (ii) providing

---

**Prompt Template**

*System*

You are an expert in tabular data analysis. You are given a user query and a set of tables. Find the query answers.

*User*

\# The *query* is the question you need to answer
\# The set of tables are the source of information you can retrieve to help you answer the given query.

Now, follow the provided information and instructions below.

\# Step One: Find most relevant tables to answer the query.
1. Read the query and the tables carefully.
2. Given the query, figure out and find the most relevant tables (normally 1-3 tables) from the set of table nodes to answer the query.
3. The inter-relationship among each node is also provided in the graph-related information, which will be provided later.
4. Once you have identified the relevant tables, follow the step two to answer the query.

\#\# The query is : *<query>*

\#\# The retrieved tables are:
*<table1>…</table1>, <table2>…</table2>*
———— *In Html Format*
\#\# Graph Related Informtion
{"source_node": "Table 1", "target_node": "Table 2", "relationship": {"type": "similarity", "score": 0.674}}
…

\# Step Two: Answer the query based on the retrieved tables
1. The detailed instruction for this tasks type is:
*Use the retrieved most relevant tables to verify whether the provided claim/query are true or false. Work through the problem step by step, and then return 0 if it's false, or 1 if it's true. Only return 0 or 1 without any other information.*
———— *Example Instruction for TFV*

\#Step Three: Here we provide output instructions that you MUST strictly follow.
1. You MUST think step by step via the chain-of-thought for the given task and then give a final answer.
2. Your output MUST conclude two components: the chain-of-thought (CoT) steps to reach the final answer and the final answer.
3. For the CoT component, you MUST enclose your reasoning between <reasoning> and </reasoning> tags.
4. For the final answer component, you MUST enclose your reasoning between <answer> and </answer> tags.

Here are few-shot examples to demonstrate the final answer component format:
*<Example1> <Example2> …*

5. If you try your best but still cannot find the answer from both the given table sources and your pretrained knowledge, then output your thinking steps and the final answer using <answer>NA</answer> to indicate that the answer can not be answered.

\# Now Output Your response below:

*Assistant*
…

Figure 4: Prompt Template for downstream LLMs tabular reasoning.

instructions for table retrieval, and (iii) offering specific guidance for long CoT output generation. An illustration of our overall prompt template is presented in Figure 4.

Table 8: Downstream results of LLMs tabular reasoning using the baseline method of TAPAS.

| Models | TFV | | | Single-hop TQA | | | | | | Multi-hop TQA | | | | | |
|---|---|---|---|---|---|---|---|---|---|---|---|---|---|---|---|
| | EM@10 | EM@20 | EM@50 | EM@10 | F1@10 | EM@20 | F1@20 | EM@50 | F1@50 | EM@10 | F1@10 | EM@20 | F1@20 | EM@50 | F1@50 |
| Phi-3.5-mini | 19.6 | 41.4 | 39.7 | 22.9 | 24.5 | 23.5 | 25.1 | 22.3 | 23.5 | 11.1 | 11.6 | 10.9 | 11.3 | 12.2 | 12.8 |
| LLaMA-3.2-3B | 36.3 | 43.5 | 38.9 | 16.8 | 17.3 | 20.0 | 20.5 | 20.7 | 21.4 | 9.9 | 12.9 | 14.1 | 14.9 | 11.7 | 12.2 |
| Qwen-2.5-7B | 41.8 | 48.5 | 42.0 | 27.5 | 31.0 | 27.5 | 29.1 | 32.6 | 25.3 | 22.0 | 24.6 | 21.7 | 24.3 | 25.4 | 26.1 |
| LLaMA-3.1-8B | 42.5 | 47.2 | 45.3 | 29.3 | 30.6 | 28.9 | 30.5 | 28.0 | 29.1 | 22.1 | 23.0 | 23.4 | 23.9 | 25.8 | 24.3 |
| LLaMA-3.1-70B | 45.2 | 50.7 | 57.8 | 37.8 | 40.1 | 39.0 | 41.5 | 42.3 | 44.2 | 28.2 | 27.7 | 29.3 | 28.5 | 32.0 | 32.3 |
| Claude-3.5-Sonnet | 47.3 | 56.4 | 61.2 | 43.9 | 48.1 | 47.2 | 49.1 | 47.0 | 47.8 | 39.8 | 40.3 | 41.9 | 42.5 | 39.3 | 39.8 |
| GPT-4o-mini | 39.8 | 46.8 | 51.8 | 34.9 | 35.1 | 34.1 | 34.5 | 36.5 | 39.2 | 33.0 | 34.1 | 31.2 | 32.0 | 32.4 | 33.5 |
| GPT-4o | 46.3 | 57.0 | 60.9 | 42.4 | 45.6 | 43.3 | 46.0 | 45.2 | 49.1 | 37.0 | 38.1 | 40.1 | 41.4 | 44.0 | 45.2 |

# B  MULTITABLEQA DETAILS

## B.1  PRELIMINARY ANALYSIS

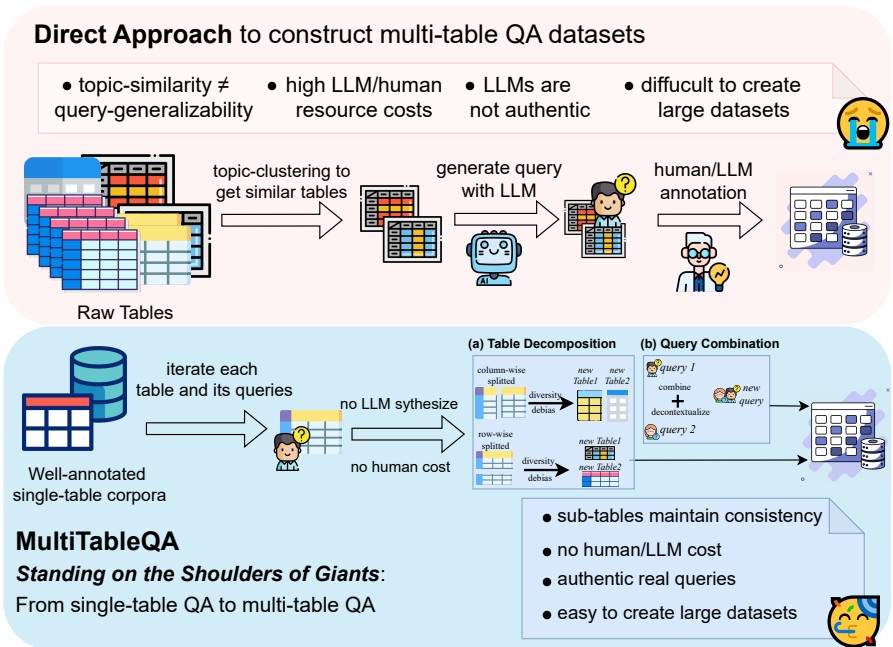

Figure 5: Illustration of the direct multi-table dataset construction approach and the MultiTableQA construction pipeline. We employ table decomposition and query combination techniques to convert real-world single-table QA tasks into multi-table QA settings.

The primary challenge in constructing the dataset lies in collecting multi-table data along with corresponding queries. A straightforward approach involves semantically clustering similar tables from single-table resources, such as Spider or Wiki-Table, into joint table sets, and then employing human experts or automated annotators (e.g., LLMs) to design queries based on these clusters.

As illustrated in Figure 5 (a), the common dataset construction method has several drawbacks: *(i) Sparsity of similar-topic table sets:* In our preliminary experiments, we clustered tables using either primary/foreign key relationships or semantic cosine similarity of table schemas. However, even within the same topical category, tables often exhibit substantial heterogeneity. For instance, under the clustered category of "NFL Games", one table's content is about "player information" and another is about "NFL team matches & schedules". This intrinsic sparsity complicates further topic refinement and downstream query annotation. *(ii) High Resource Costs:* Annotating queries that require reasoning across multiple tables with human experts or LLMs is highly resource-intensive, limiting the scalability of constructed datasets. *(iii) Auto-annotation Bias:* Relying on auto-annotators (e.g., LLMs) for multi-table queries often introduces bias, as both the queries and their ground-truth

labels are model-generated. This divergence may compromise the realism of RAG settings, which are based on authentic user queries and source data.

To overcome these challenges, we reframe the data construction process by **decomposing source tables** and **combining queries**. As illustrated in Figure 5 (b), our strategy guarantees that the resulting sub-tables, all derived from a single root table, maintain an intrinsic relationship, while the original queries now necessitate multi-hop reasoning across these sub-tables.

## B.2 TASK TYPE DEFINITION

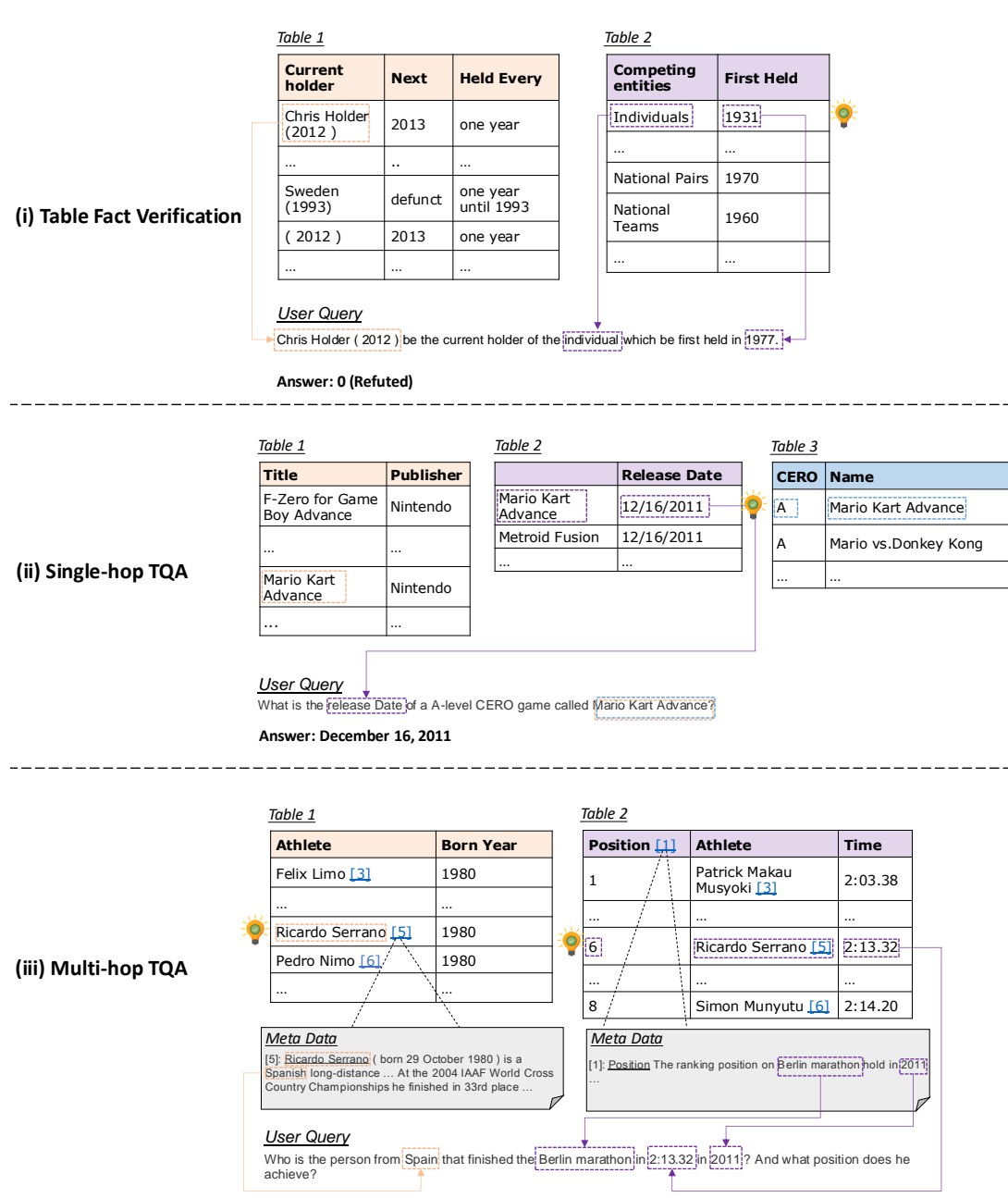

Figure 6: Demonstration on three different task types in MultiTableQA.

We give detailed explanations of the three task types in MultiTableQA. Figure 6 provides a concrete example for each task.

- **Table-based Fact Verification** determines whether a user-provided claim is supported or refuted based on the given tabular data. In our benchmark, we label an entailed (supported) claim as "1" and a refuted claim as "0", depending on the evidence found within the tables.

- **Single-hop Table Question Answering** focuses on questions that can be answered using information from a single table cell. However, identifying the correct cell often requires the LLMs to reason across multiple tables and recognize connections between different pieces of tabular information, as illustrated in the example figure.

- **Multi-hop Table Question Answering** addresses questions that require reasoning over multiple cells, often across different rows, columns, or tables. The final answer typically consists of a list of strings aggregated from multiple relevant entries in the tables.

## C  EXPERIMENT SETUP

### C.1  BASELINE METHODS DESCRIPTION

The detailed baseline method descriptions for each category are listed below:

- **Table Retrieval.** This category includes methods that leverage table-aware retrievers—often fine-tuned on structured data—to identify and rank the most relevant tables given a query. These approaches focus on selecting either a single best-matching table or aggregating information across multiple retrieved tables. In our experiments, we choose common table retrieval methods including DTR (Herzig et al., 2021), Table-Contriever (Izacard et al., 2021), Table-E5 (Wang et al., 2022b) and Table-LLaMA (Zhang et al., 2023a) as our baseline methods.

- **RAG-based.** The RAG-based methods normally integrate a retriever with a generator. The retriever identifies relevant tables from a large corpus, which are then passed to the generator to produce context-aware outputs. This paradigm enhances generation quality by grounding the response in retrieved evidence, making it particularly effective for knowledge-intensive tasks. In our experiments, we utilize In-context RALM (Ram et al., 2023) (Abbreviated as RALM) and ColBERTv2 (Santhanam et al., 2021) (Abbreviated as ColBERT) as our baseline methods.

- **Table-to-Graph Representation.** The table-to-graph representation represents different node feature representation approaches, as compared to our method described in Section 3.1. Specifically, we compare with single feature extraction methods (e.g., semantic, structural, or heuristic features alone), tabular representation learning models such as TAPAS (Herzig et al., 2020) and TaBERT (Yin et al., 2020), as well as table summarization methods such as Lattice (Wang et al., 2022a).

- **Table Prompting Methods.** These approaches encode tabular data into natural language prompts that are fed into LLMs. By linearizing the table content or formatting it into structured textual representations, these methods enable LLMs to effectively reason over tabular inputs. In our experiments, we choose TAP4LLM (Sui et al., 2023) as our baseline method as it includes multiple plug-and-play table sampling, augmentation, and packing methods inside.

### C.2  IMPLEMENTATION DETAILS

**Model Settings.**  We conduct a comprehensive evaluation of downstream tabular reasoning using both open- and closed-source LLMs. Specifically, for closed-source models, we employ Claude-3-5-Sonnet-2024-10-22 (Anthropic, 2024) (abbreviated as Claude-3.5-Sonnet), GPT-4o-2024-08-06 (Hurst et al., 2024) (abbreviated as GPT-4o), and GPT-4o-mini-2024-07-18 (Hurst et al., 2024) (abbreviated as GPT-4o-mini). For open-source models, we utilize the LLaMA3 families (Grattafiori et al., 2024) including LLaMA-3.1-8B-Instruct, LLaMA-3.1-70B-Instruct, and LLaMA-3.2-3B-Instruct, Phi-3.5-mini-Instruct (Abdin et al., 2024), and Qwen-2.5-7B-Instruct (Yang et al., 2024). In our experiment settings, we omit all "Instruct" in the model names for brevity. For the model parameter setups, we set the temperature to 0.1, max output tokens to 4096, and top-p to 0.95.

**Baseline Implementation.** For the Table Retrieval and RAG-based baselines, we first linearize the tables using the *Table Linearization* procedure described in Section 3.1. We then apply the respective retrievers to the resulting table sequences, retrieving the top 10, 20, and 50 relevant tables to serve as the final input for downstream LLMs. For all Table-to-Graph baseline methods, we encode each table into an embedding representation using the respective approach, compute a similarity matrix from these embeddings, and then apply personalized PageRank to retrieve the final set of table nodes, as described in our method. For all other baselines, we strictly follow the publicly available code implementations corresponding to each method. For the downstream LLMs reasoning, we use the same graph-aware prompt settings as our method to ensure fair comparison.

## D  ADDITIONAL EXPERIMENTS

### D.1  DOWNSTREAM LLMS TABULAR REASONING

Table 3 presents the overall downstream performance of T-RAG. For comparison, we also report the downstream results of the strongest baselines, including Table-E5 (as shown in Table 7) and TAPAS (as shown in Table 8). From the results, we observe that the superior retrieval capability of T-RAG yields notable performance gains on downstream LLMs compared to other table retrieval and table-to-graph representation methods. However, we also find that increasing the number of final retrieved tables can sometimes lead to performance degradation, indicating that improved retrieval metrics do not always correlate with enhanced downstream performance. We discuss these observations in detail in the subsequent ablation and sensitivity studies.

### D.2  ABLATION STUDY

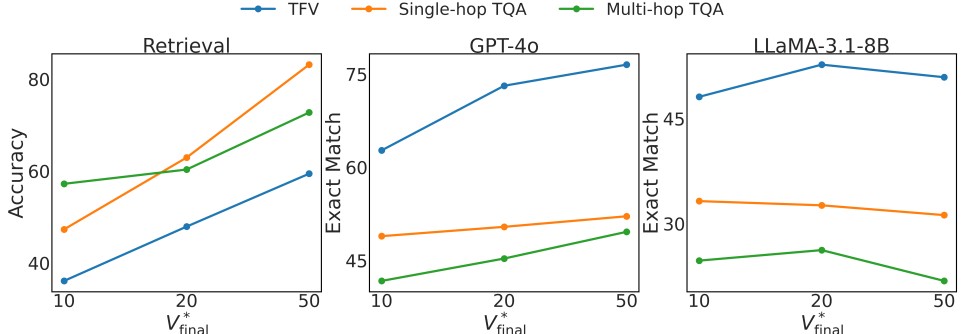

Figure 7: Ablation study on $\mathcal{V}_{\text{final}}^*$

Table 9: Comprison of T-RAG with other table prompt baseline. We evaluate on GPT-4o and report the EM@50 results."G" Stands for the Graph Information Insertion part, "H" stands for the Hierarchical Long CoT Generation (Generated CoT part between <reasoning> and <reasoning/>).

| Methods | TFV | Single-hop TQA | Multi-hop TQA |
|---------|-----|----------------|---------------|
| TAP4LLM | 61.4 | 53.8 | 46.9 |
| **T-RAG** | 66.5 | 56.6 | 50.9 |
| *w/o* G | 65.3 | 55.9 | 49.4 |
| *w/o* H | 58.4 | 43.7 | 38.6 |

**Graph-aware Prompting.** To demonstrate the effectiveness of our graph-aware prompting, we compare our approach against TAP4LLM and variants of our method that exclude the Graph Information Insertion and hierarchical Long CoT generation components, as shown in Table 9. The results show that each component contributes to preserving table interconnectivity and providing clear guidance for multi-step reasoning on the challenging cross-table retrieval tasks.

**Choice of $\mathcal{V}^*_{\text{final}}$.** In Tables 2 and 3, we evaluate the performance of T-RAG using different numbers of final retrieved tables, denoted as $\mathcal{V}^*_{\text{final}}$. For better demonstration, Figure 7 illustrates the retrieval and downstream tabular reasoning performance for GPT-4o and LLaMA-3.1-8B. We observe that retrieval accuracy and recall consistently increase as the number of retrieved tables grows. For downstream tasks, GPT-4o demonstrates enhanced performance with a larger number of tables, whereas LLaMA-3.1-8B exhibits performance degradation when the number of tables increases from 20 to 50. A likely explanation is that a higher number of retrieved tables leads to longer input prompts. Models with larger parameter sizes can effectively handle extended contexts and extract useful information from additional tables, while smaller models may struggle with lengthy prompts and thus fail to fully leverage the retrieved information.

Table 10: Ablation study on hyperparameters settings in coarse-grained multi-way retrieval. We compare the accuracy and report the average number of retrieved tables in the optimal cluster. Hyperparameters adapted in our implementation are underlined.

| Hyperparameters | TFV | Single-hop TQA | Multi-hop TQA | Avg. Tables |
|---|---|---|---|---|
| $K = 3$ | 96.1 | 98.3 | 93.6 | 9027 |
| $K = 5$ | 94.1 | 96.5 | 89.4 | 6438 |
| $\underline{K = 10}$ | 84.2 | 95.9 | 87.6 | 2426 |
| $K = 20$ | 71.3 | 85.4 | 70.2 | 2163 |
| $k = 50$ | 77.5 | 89.2 | 72.8 | 5569 |
| $\underline{k = 100}$ | 84.2 | 95.9 | 87.6 | 2426 |
| $k = 150$ | 86.2 | 96.2 | 88.2 | 4799 |
| $k = 200$ | 83.7 | 89.3 | 74.4 | 5628 |

### D.3 PARAMETER STUDY

**Hyperparameters on Multi-way Retrieval.** We investigate the impact of the number of clusters $K$ and the number of top-$k$ typical nodes during the coarse-grained multi-way retrieval (Section 3.2). Table 10 presents the comparison results, reporting the accuracy (i.e., the proportion of sub-tables and corresponding testing queries grouped into the optimal cluster). The experimental results reveal that using fewer clusters yields higher accuracy, but it also results in a substantial increase in the number of tables per cluster. Moreover, setting the top-$k$ typical nodes too low or too high can lead to performance degradation or overfitting, respectively. Based on these observations, we adopt $K = 10$ and $k = 100$ in our implementation.

**Hyperparameters on Subgraph Retrieval.** Figure 8 presents the retrieval performance for various settings of the PageRank sampling factor $\alpha$ and the similarity threshold $\tau$ employed during the fine-grained subgraph retrieval process (Section 3.3). We extensively tuned these hyperparameters across a wide range. Our results indicate that $\alpha = 0.85$ generally yields the best performance across all three tabular reasoning tasks, which aligns with several previous PageRank studies (Rozenshtein and Gionis, 2016; Yoon et al., 2018). However, the optimal value of $\tau$ varies depending on the task due to differences in sparsity and scalability among our constructed graphs. Consequently, we tuned $\tau$ over the range $[0, 1]$ to determine its most effective setting.

## E LLM USAGE DISCLOSURE

During the paper writing process, LLMs are utilized to formalize the writing in terms of word choice and grammar correction.

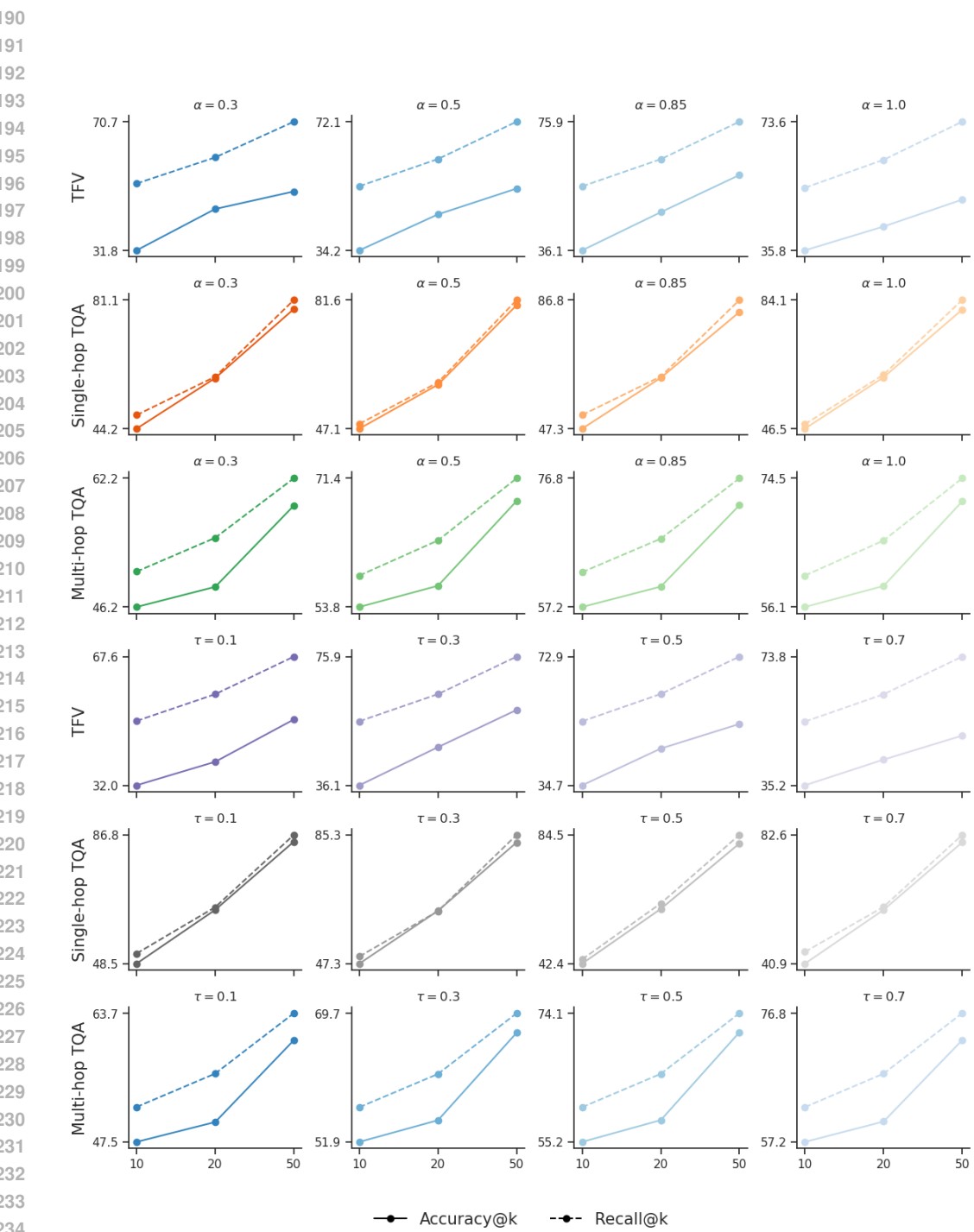

Figure 8: Ablation study on hyperparameters settings in fine-grained subgraph retrieval. For each experiment, all other hyperparameters remain fixed. We compare the retrieval accuracy and recall.

