# OpenReview forum: "RAG over Tables: Hierarchical Memory Index, Multi-Stage Retrieval, and Benchmarking"
_ICLR.cc/2026/Conference — ICLR 2026 Conference Withdrawn Submission_

### Official Review · Reviewer_JjWc · 2025-10-18

**Soundness:** 2
**Presentation:** 3
**Contribution:** 2
**Rating:** 4
**Confidence:** 4

**Summary:**

This paper presents T-RAG, a novel framework for RAG designed to answer complex questions requiring information distributed across multiple tables. The framework features three key components: a hierarchical memory index that organizes tables into a graph, a multi-stage retrieval process to efficiently filter and find relevant tables, and a graph-aware prompting method to enhance LLM reasoning. To evaluate the system, the authors also introduce MultiTableQA, a new large-scale benchmark consisting of 57,193 tables and 23,758 questions from real-world sources. Experiments show that T-RAG outperforms baseline methods in accuracy, recall, and efficiency, and improves the tabular reasoning capabilities of various LLMs.

**Strengths:**

1. The authors address the important and practical task of table retrieval and the challenging problem of multi-table question answering, as highlighted in the paper.
2. The proposed T-RAG pipeline demonstrates a significant improvement in both retrieval and table reasoning performance.

**Weaknesses:**

1. **Insufficient Motivation**: The motivation is not well-established. Numerous table question answering datasets already exist (e.g., Open-WikiTable [1], MMQA [2], RETQA [3], TANQ [4], TARGET [5]), yet the authors do not provide a detailed comparison with them, making it difficult to situate the contribution of the new dataset.
2. **Unconvincing Data Construction Method**: The "Query Combination" method for data construction is overly simplistic. I am skeptical about the difficulty of the resulting queries. A key challenge in multi-table QA is that a single user query may require the model to sequentially retrieve and link information from multiple tables, akin to a multi-table join operation in a database. However, simply using connecting words (e.g., “AND”, “Furthermore”, “Based on [previous query]”, line 339) is unlikely to produce queries that require such complex, chained retrieval. Instead, this method tends to generate queries that are compositions of independent questions, which significantly lowers the retrieval difficulty.
3. **Inadequate Baselines**: The authors propose a complex pipeline but only compare it against basic retrieval methods. The absence of comparisons with more advanced or state-of-the-art table retrieval methods, such as OpenTab [6] and MTR [1], calls the effectiveness of the T-RAG pipeline into question.

[1] Open-WikiTable: Dataset for Open Domain Question Answering with Complex Reasoning over Table

[2] MMQA: EVALUATING LLMS WITH MULTI-TABLE MULTI-HOP COMPLEX QUESTIONS

[3] RETQA: A Large-Scale Open-Domain Tabular Question Answering Dataset for Real Estate Sector

[4] TANQ: An open domain dataset of table answered questions

[5] TARGET: Benchmarking Table Retrieval for Generative Tasks

[6] OPENTAB: ADVANCING LARGE LANGUAGE MODELS AS OPEN-DOMAIN TABLE REASONERS

**Questions:**

Regarding the examples in Appendix B.2, there are no explicit connecting words in the "Table Fact Verification" and "Single-hop TQA" queries. This is confusing, as it does not seem to have been synthesized by the rule-based "Query Combination" method described. Could you please provide a more detailed explanation of how these queries were merged?

---

### Official Review · Reviewer_Htwx · 2025-10-29

**Soundness:** 2
**Presentation:** 2
**Contribution:** 2
**Rating:** 4
**Confidence:** 4

**Summary:**

This paper introduces T-RAG, a novel Retrieval-Augmented Generation (RAG) framework meticulously designed for efficient retrieval and complex reasoning over large-scale, loosely connected table corpora, a domain where traditional RAG systems struggle. The framework's core innovations include a hierarchical memory index, which models the tables as a heterogeneous hypergraph using multi-way feature representations (semantic, structural, and heuristic), and a multi-stage retrieval pipeline that efficiently narrows the search space using PageRank on a local subgraph. Furthermore, T-RAG employs a graph-aware prompting mechanism to guide Large Language Models (LLMs) through multi-step reasoning via Hierarchical Long Chain-of-Thought instructions. Validation is provided through the new, large-scale MultiTableQA benchmark, and results demonstrate T-RAG’s state-of-the-art performance in retrieval and its ability to significantly enhance the tabular question answering capabilities of various downstream LLMs, achieving an average performance improvement of 11.2%.

**Strengths:**

1. Strong Methodological Integration for Tabular Data: The framework successfully addresses the complexity of tabular data—which involves text, schema, and structure—by modeling the table corpus as a heterogeneous hypergraph and utilizing multi-way feature extraction (semantic, structural, and heuristic). This integrated approach effectively captures diverse table relationships, achieving high recall and enabling efficient hierarchical indexing.


2. Superior Retrieval Performance: T-RAG consistently achieves state-of-the-art performance in retrieval accuracy, recall, and runtime efficiency when compared against major paradigms, including dedicated table retrieval methods, general RAG methods, and table-to-graph representation learning techniques.


3. Significant Downstream LLM Enhancement: The superior retrieval capability directly translates to consistent and measurable gains in downstream question answering. T-RAG demonstrates robust efficacy across various LLMs (e.g., GPT-4o, Claude-3.5-Sonnet), yielding an impressive average performance improvement of 11.2% in tabular reasoning compared to the strongest baseline methods.

**Weaknesses:**

W1: Computational Bottleneck in Fine-grained Retrieval: While coarse-grained filtering is highly efficient, the Fine-grained retrieval step using Iterative Personalized PageRank (PPR) incurs significant latency, making the overall end-to-end efficiency only comparable to, or slightly higher than, strong baselines for large tasks like Table-based Fact Verification (TFV).


W2: Performance Degradation in Smaller LLMs: Smaller LLM backbones, such as LLaMA-3.1-8B, exhibit a performance degradation when provided with a larger number of retrieved tables (e.g., from 20 to 50), suggesting they struggle with contextual noise or exceeding their effective context window limits.


W3:High Sensitivity to Hyperparameter Tuning: The performance of the coarse-grained multi-way retrieval stage is highly dependent on the choice of hyperparameters, such as the number of clusters ($K$) and typical nodes ($k$), requiring careful, corpus-specific tuning to maintain efficiency without compromising accuracy.

W4: Artificial Structure in the Benchmark Data: The MultiTableQA benchmark relies on synthesized relational structures created by decomposing single root tables, which risks biasing results toward models recognizing decomposition artifacts rather than complex, implicit relationships found in organically sourced multi-table data.

W5: Heavy Reliance on Structured Prompting: The success of the generative stage is critically dependent on the Hierarchical Long Chain-of-Thought (CoT) prompting mechanism; empirical evidence shows that removing this explicit structural guidance causes a massive degradation in downstream LLM performance.

W6: Context Distillation Capacity Limitation: The T-RAG framework introduces a "retrieval volume paradox" where improved retrieval metrics (accuracy/recall) do not guarantee downstream gains, as the framework is fundamentally limited by the LLM’s capacity to distill necessary information from an increased volume of retrieved context.

**Questions:**

same as weakness

---

### Official Review · Reviewer_vJHz · 2025-10-31

**Soundness:** 2
**Presentation:** 2
**Contribution:** 1
**Rating:** 2
**Confidence:** 4

**Summary:**

The authors propose T-RAG, a retrieval-augmented generation (RAG) framework that retrieves relevant knowledge from tables to improve the accuracy of large language model (LLM) inference, enabling effective multi-table retrieval across large-scale table corpora. The framework consists of three key components: a hierarchical memory index that clusters tables with similar properties for efficient retrieval, a multi-stage retrieval process that identifies query-relevant clusters and then refines table candidates based on their relevance to the query, and a prompting method that captures the relationships among tables in the same cluster and delivers it to the LLM. Furthermore, the authors introduce MultiTableQA, a benchmark for multi-table question answering, to evaluate cross-table retrieval and reasoning performance. Experimental results show that T-RAG outperforms baseline models such as Table-E5 and TAPAS by an average margin of 11.2%.

**Strengths:**

S1. The authors address an important challenge in real-world question answering: retrieving relevant knowledge across large-scale table corpora.

S2. The authors introduce a new benchmark, MultiTableQA,  designed to evaluate and advance research on multi-table question answering.

**Weaknesses:**

W1. The novelty of T-RAG is unclear, as the framework appears to combine existing methods rather than making a fundamentally new contribution.

W1.1 The authors regard the hierarchical memory index construction using clustering as a hypergraph-based approach; however, this has already been well-established in prior research [1]. Moreover, the proposed index-building method applies K-Means clustering, a commonly adopted approach in retrieval studies [2, 3], directly to table feature vectors without introducing meaningful adaptations.

W1.2 The combined use of dense and sparse vectors has been widely explored in existing work [4, 5], making this aspect appear derivative rather than innovative.

W1.3 Similarly, the use of personalized PageRank to score vectors from selected clusters closely mirrors techniques already employed in prior graph-based RAG studies [6, 7, 8]. These overlaps render T-RAG's contributions more incremental than fundamentally novel

[1] Zhou, Dengyong, Jiayuan Huang, and Bernhard Schölkopf. "Learning with hypergraphs: Clustering, classification, and embedding." Advances in neural information processing systems 19 (2006).

[2] Chen, Qi, et al. "Spann: Highly-efficient billion-scale approximate nearest neighborhood search." Advances in Neural Information Processing Systems 34 (2021): 5199-5212.

[3] Santhanam, Keshav, et al. "PLAID: an efficient engine for late interaction retrieval." Proceedings of the 31st ACM International Conference on Information & Knowledge Management. 2022.

[4] Zhuang, Shengyao, et al. "PromptReps: Prompting Large Language Models to Generate Dense and Sparse Representations for Zero-Shot Document Retrieval." Proceedings of the 2024 Conference on Empirical Methods in Natural Language Processing. 2024.

[5] Karpukhin, Vladimir, et al. "Dense Passage Retrieval for Open-Domain Question Answering." Proceedings of the 2020 Conference on Empirical Methods in Natural Language Processing (EMNLP). 2020.

[6] Jimenez Gutierrez, Bernal, et al. "Hipporag: Neurobiologically inspired long-term memory for large language models." Advances in Neural Information Processing Systems 37 (2024): 59532-59569.

[7] Gutiérrez, Bernal Jiménez, et al. "From RAG to Memory: Non-Parametric Continual Learning for Large Language Models." Forty-second International Conference on Machine Learning.

[8] Wang, Jingjin. "PropRAG: Guiding Retrieval with Beam Search over Proposition Paths." arXiv preprint arXiv:2504.18070 (2025).

W2. The paper lacks sufficient comparison with existing models and benchmarks.
W2.1 The proposed method is not compared against state-of-the-art graph-RAG approaches [6, 7, 8] and table retrieval methods [9, 10, 11]; the authors should clearly identify the key differences and include explicit experimental comparisons --  both in terms of retrieval accuracy and end-to-end QA performance -- to substantiate claims of superiority.
W2.2 The evaluation considers only a single prompting baseline for generation, omitting several relevant graph-aware prompting techniques, such as the hard prompt template in GRAG [12], knowledge graph-based prompting in KnowGPT [13], and prompt templates in SubgraphRAG [14]. Incorporating these baselines and conducting experiments on end-to-end QA accuracy would significantly increase the empirical support for the proposed framework.
W2.3 The authors claim MultiTableQA is the first benchmark for multi-table QA, but prior work has introduced similar benchmarks [15, 16]. The paper would benefit from a more detailed comparison highlighting specific distinctions  -- such as dataset scale or complexity of queries-- to better justify MultiTableQA's novelty and contributions.
[9] Lin, Weizhe, et al. "Li-rage: Late interaction retrieval augmented generation with explicit signals for open-domain table question answering." Proceedings of the 61st Annual Meeting of the Association for Computational Linguistics (Volume 2: Short Papers). 2023.
[10] Chen, Peter Baile, Yi Zhang, and Dan Roth. "Is Table Retrieval a Solved Problem? Exploring Join-Aware Multi-Table Retrieval." Proceedings of the 62nd Annual Meeting of the Association for Computational Linguistics (Volume 1: Long Papers). 2024.
[11] Wu, Jian, et al. "MMQA: Evaluating LLMs with multi-table multi-hop complex questions." The Thirteenth International Conference on Learning Representations. 2025.
[12] Hu, Lei, et al. GRAG: Graph Retrieval-Augmented Generation. In Findings of the Association for Computational Linguistics: NAACL 2025, pages 4145–4157, Albuquerque, New Mexico. Association for Computational Linguistics.
[13] Zhang, Qinggang, et al. "Knowgpt: Knowledge graph based prompting for large language models." Advances in Neural Information Processing Systems 37 (2024): 6052-6080.
[14] Li, Mufei, Siqi Miao, and Pan Li. "Simple is Effective: The Roles of Graphs and Large Language Models in Knowledge-Graph-Based Retrieval-Augmented Generation." The Thirteenth International Conference on Learning Representations.
[15] Wu, Jian, et al. "MMQA: Evaluating LLMs with multi-table multi-hop complex questions." The Thirteenth International Conference on Learning Representations. 2025.
[16] Qiu, Zipeng, et al. "Tqa-bench: Evaluating llms for multi-table question answering with scalable context and symbolic extension." arXiv preprint arXiv:2411.19504 (2024).

W3. The paper would benefit from clearer alignment between its methodological choices and the corresponding technical motivations.

W3.1 In generating embeddings for tables, the proposed approach excludes cell values and relies solely on the schema information. However, the rationale for this design choice is not discussed. The authors should elaborate on the underlying motivation -- such as potential trade-offs between representation efficiency and semantic richness -- to better justify this decision.

W3.2 The motivation for employing the Typical Node Selection is insufficiently explained. Although this method increases computational overhead compared to simpler alternatives like direct centroid-based similarity, the paper does not articulate the specific advantages. To address this, the authors should include experimental evidence comparing accuracy and efficiency to show the empirical value of this design choice.

**Questions:**

Please refer to W1, W2, and W3.

---

### Official Review · Reviewer_cGTr · 2025-10-31

**Soundness:** 2
**Presentation:** 2
**Contribution:** 1
**Rating:** 2
**Confidence:** 4

**Summary:**

This paper proposes T-RAG, a table-corpora-aware Retrieval-Augmented Generation framework designed to enhance LLMs in retrieving and reasoning over tabular data. T-RAG introduces a hierarchical memory index, multi-stage retrieval, and graph-aware prompting to effectively handle intra- and inter-table knowledge. Additionally, the authors present MultiTableQA, a large-scale benchmark with over 57,000 tables and 23,000 questions, on which T-RAG demonstrates superior performance in accuracy, recall, and efficiency compared to existing baselines.

**Strengths:**

The writing and presentation is clear.

**Weaknesses:**

**Majors**

1. I do not agree with the statement in Lines 51–52 that the information format in tables is more complex than pure text. The reasons are as follows:
  (1) Tables present information in a clear, well-organized, and structured manner. This is precisely why they are widely used in research papers.
(2) Tables convey information accurately and concisely, especially when accompanied by clear captions. In contrast, describing the same information entirely in text would require lengthy and potentially confusing sentences.

2. The proposed T-RAG method lacks novelty and appears to be more of an integration of several existing techniques rather than a fundamentally new approach. As stated by the authors in Section 1, both the hierarchical design (Lines 60–61) and the graph-based design (Lines 39–41) are widely adopted techniques. Therefore, it remains unclear what novel design elements or key insights are specifically introduced by T-RAG.

3. The advantages and necessity of constructing the MultiTableQA benchmark are unclear. It is not well justified why such a new benchmark is needed, given the significant effort required to build it. The authors should clarify what specific gaps in existing table QA or multi-step QA benchmarks motivate the creation of MultiTableQA, and what unique advantages or new challenges it introduces beyond existing datasets.

4. The main contribution of this paper is the proposed T-RAG method. However, the experiments are primarily conducted on the newly introduced MultiTableQA benchmark. Although the authors claim that T-RAG has strong generalization potential, evaluating it solely on the Spider dataset is clearly insufficient to support this claim.

---

**Minors**

5. The tasks in MultiTableQA bench are limited within 3 types. It is just a combination of existing benmarks.

6. The performance and efficiency are not very good.

**Questions:**

Please see weaknesses.

---

### Note · Authors · 2026-01-06

I have read and agree with the venue's withdrawal policy on behalf of myself and my co-authors.